Review

# Transducing compressive forces into cellular outputs in cancer and beyond

Céline Schmitter[1,2,3] ⓘ, Mickaël Di-Luoffo[1,2], Julie Guillermet-Guibert[1,2] ⓘ

**In living organisms, cells sense mechanical forces (shearing, tensile, and compressive) and respond to those physical cues through a process called mechanotransduction. This process includes the simultaneous activation of biochemical signaling pathways. Recent studies mostly on human cells revealed that compressive forces selectively modulate a wide range of cell behavior, both in compressed and in neighboring less compressed cells. Besides participating in tissue homeostasis such as bone healing, compression is also involved in pathologies, including intervertebral disc degeneration or solid cancers. In this review, we will summarize the current scattered knowledge of compression-induced cell signaling pathways and their subsequent cellular outputs, both in physiological and pathological conditions, such as solid cancers.**

## Introduction

Shear, tension, and compression are ubiquitous mechanical forces exerting physiological responses on cells. Shearing force corresponds to the application of a force that is parallel to cell surface; the force applied to cells is perpendicular and directed away from cell surface for tensile force, and perpendicular and directed toward cell surface for compressive force (Fig 1A). Mechanical forces are sensed by mechanosensors that activate biochemical effectors in signaling pathways (see Glossary). This process is called mechanotransduction (Di-Luoffo et al, 2021). Among them, signaling pathways and cellular responses induced by compressive forces are so far the least understood mechanism. We know that shearing and tensile forces lead to different mechanotransduction signaling, especially leading in the activation of different PI3K classes of enzymes and PI3K isoforms (reviewed in Di-Luoffo et al [2021]). Similarly, this difference in term of signal activation patterns is found between stretching and compressive forces applied on cells (Haudenschild et al, 2009; Takemoto et al, 2015; Nordgaard et al, 2022). This differential pathway activation has physiological implications: healthy cells such as osteoblast precursors and periodontal ligament fibroblasts produce and secrete different proteins depending on which mechanical stress they undergo (He et al, 2004; Zhong et al, 2013; Takemoto et al, 2015). To better understand the importance of mechanical forces in biological processes, there is thus a need to discriminate the selective contribution of compressive forces in activating biochemical pathways. Biological processes in vivo are subjected simultaneously to all three types of mechanical forces; disentangling the relative contribution of each physical force in cell processes is thus a complex task. To model in vitro the application of compressive forces to mammalian cells in 2D or in 3D, different methods are available. The use of those methodological approaches is expanding in the cell and tumor biology fields, described in detail in Fig 1B–D. Here, we reviewed the cellular effects of mechanical load (Fig 1B), variation in osmotic and interstitial fluid pressure because of the accumulation of hydrophilic hyaluronic acid in extracellular matrix (Fig 1C) and growth pressure in a confined environment such as rigid matrix (Fig 1D and E), all situations that mimic in vivo settings.

Sensing of compressive forces occurs at various locations in cells (Fig 2A). Mechanotransduction happens in the plasma membrane or in the actin cortical cortex. Nucleus deformation induces biochemical pathways simultaneously or in a sequential manner. Molecular or organelle crowding in cytoplasm participates in the sensing of compressive forces (Guo et al, 2017). Next, compressive forces induce different cellular outputs, ranging from proliferation to cell death (Li et al, 2017; Boyle et al, 2020; Kang et al, 2020; Lin et al, 2021) (Fig 2A–C). The direct application of compressive forces promotes or reduces cell proliferation, survival, and differentiation; it promotes cytoskeleton remodeling, cell motility, and cell migration; it controls cell metabolism. All these cell processes participate to tissue homeostasis. Compressive forces also act indirectly via autocrine or paracrine signaling (Fig 2B and C). Paracrine and autocrine action involves the regulation of secreted cytokines, chemokines (Schreivogel et al, 2019), matrix components (Wright et al, 1997; He et al, 2004; Chowdhury et al, 2006; Fitzgerald et al, 2008; Zhong et al, 2013; Liu & Lee, 2014; Takemoto et al, 2015; Luo et al, 2022), metabolites (McCutchen et al, 2017), and the control

[1]CRCT, Université de Toulouse, Inserm, CNRS, Université Toulouse-III Paul Sabatier, Centre de Recherches en Cancérologie de Toulouse, Toulouse, France [2]Labex Toucan, Toulouse, France [3]Master de Biologie, Ecole Normale Supérieure de Lyon, Université Claude Bernard Lyon 1, Université de Lyon, Lyon, France

Correspondence: julie.guillermet@inserm.fr

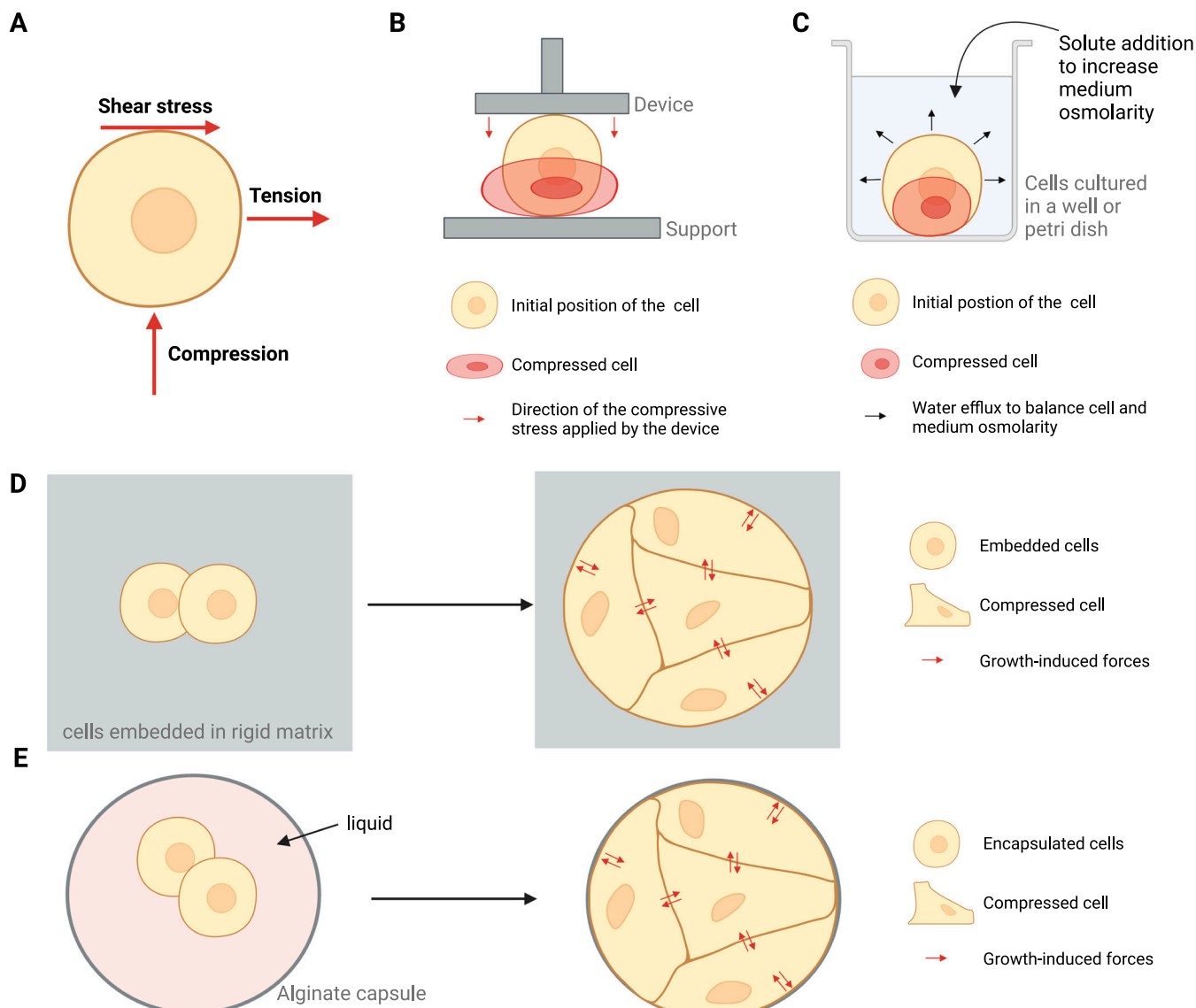

**Figure 1. Experimental designs used to mimic in vitro compressive force generation in 2D and 3D cell cultures.**
**(A)** Scheme representing how compression, tension, and shearing forces are applied to the cell membrane. Red arrows represent the direction of the corresponding force applied to the cell. Several methods are used in vitro to reproduce compressive forces that are found in vivo. **(B)** The compressive force applied can be static (i.e., applied one time during a defined period) or cyclic (i.e., applied in cycles of compression) using the following methods: (B) compression induced by a piston filled with an adjustable weight (static) or by a transmembrane pressure device using compressed gas to press a piston towards the cells (cyclic). Here, only 2D compression is shown; however, compression of 3D multicellular structures can be achieved by those methods. **(C)** Compression induced by hyperosmotic shock. For example, addition of polyethylene glycol-300 to cell media increases cell media osmolarity resulting in water efflux from cell to media to equilibrate osmolarity and thus causing cell compression. **(D)** Confined growth of cells or of multicellular structures in rigid matrix (>1 kPa) mimics the buildup of growth-induced compressive forces that are mostly found during tumor development. Refinement of those approaches can be achieved by using gels that can relax and mimic the viscoelastic properties of extracellular matrix. The rigid matrix can also be functionalized to mimic ECM composition and the concomitant biochemical activation through protein/protein interaction of confined cells. **(E)** Confinement of cells in rigid capsules. They can grow as a mass inside the capsule or line the alginate capsule; the latter mimics the formation of simple epithelium. The described experimental devices were used in the following references (Tse et al, 2012; Guo et al, 2017; Kalli & Stylianopoulos, 2018; Kim et al, 2019; Boyle et al, 2020; Kang et al, 2020; Nia et al, 2020; Li et al, 2021b; Ge et al, 2021).

of membrane receptor recycling (Baschieri et al, 2020), and cell/cell adhesion (Park & Tschumperlin, 2009; Eisenhoffer et al, 2012; Delarue et al, 2017; Massey et al, 2020; Di Meglio et al, 2022).

Physical compressive forces thus control various biological processes but are also involved in pathologies in humans. In some cases, excessive compression forces lead to pathologies. It is the case of intervertebral disc degeneration (Pauly et al, 2001). During solid cancer development, the tumor growth leads to generation of compressive forces; this phenomenon modifies the tumor and microenvironment cell behavior in return (Kim et al, 2017, 2019; Morikura & Miyata, 2019). It is so far unclear whether normal, cancer, and tumor microenvironment cells respond differently to

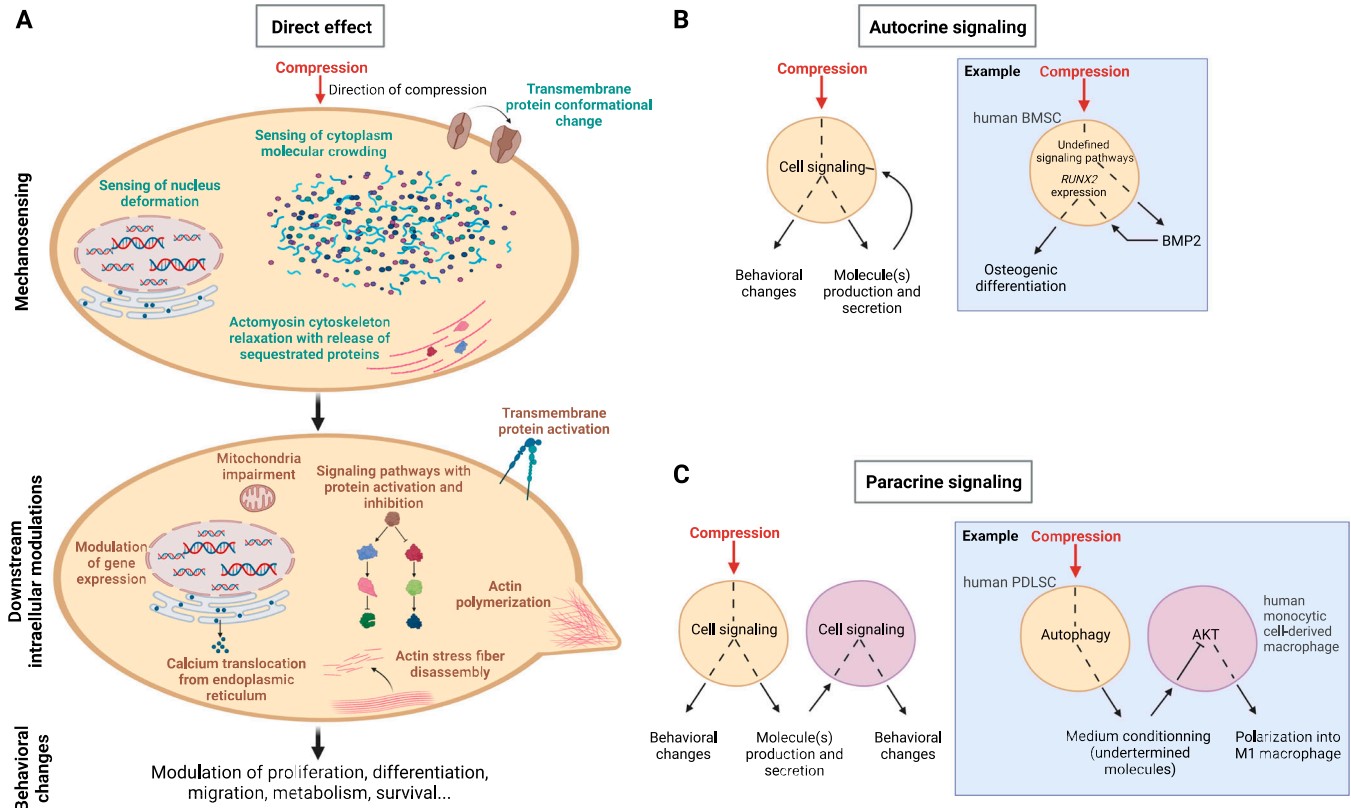

**Figure 2. General overview of compression mechanotransduction in human cells.**
**(A)** Direct compressive effect on the cell. Supposed mechanosensors and downstream modulations are displayed. This list is not exhaustive (Wright et al, 1997; Park & Tschumperlin, 2009; Eisenhoffer et al, 2012; Liu & Lee, 2014; Chronopoulos et al, 2020; Lomakin et al, 2020; Massey et al, 2020; Park et al, 2020). **(B)** Compressive stress-induced autocrine signaling. **(C)** Compressive stress-induced paracrine signaling.

compressive forces; we will review the recent data that concur to demonstrate that this response is actively promoting cancer development, progression, and resistance to treatment.

In summary, this review synthetizes by which cell signaling pathways' compression induces cellular phenotypes in mammalian cells. We will also discuss the importance of compression in health and disease with a particular focus on cancer.

## Cellular outputs modulated by compression in tissue homeostasis

Homeostatic control of cell numbers in tissues: compression increases cell proliferation of some mesenchymal cells and decreases cell proliferation or extrudes cells in most epithelia.

Cell context determines whether compression promotes or represses cell proliferation. Static (i.e., applied one time during a defined period) or cyclic (i.e., applied in cycles of compression) compressive forces (in 10 of kPa range) increase proliferation of some mesenchymal cells in a direct manner. Compression increases proliferation of rat bone marrow-derived mesenchymal stem cells (rat BMSCs) and rat chondrocytes (Ren et al, 2011; Wang et al, 2013; Boyle et al, 2020). Compression on chondrocytes triggers $Ca^{2+}$ signaling by activating mechanosensitive ion channels such as

Piezo-1 and Piezo-2, two distinct members of stretch-activated channel (SAC) family (Coste et al, 2010; Han et al, 2012; Liu & Lee, 2014). Consequently, transmembrane proteins such as $\alpha5\beta1$ integrin are activated (Wright et al, 1997; Chowdhury et al, 2006; Raizman et al, 2010; Liu & Lee, 2014). Proliferative signal is dependent on RHOA and ROCK signaling, and BMP signaling and SRC-induced MAPK/ERK pathway (Ren et al, 2011; Wang et al, 2013; Boyle et al, 2020). In response to compression, an interplay between several pathways is occurring, as an inhibition of a single signal node is not sufficient to fully block the compression-induced proliferation.

In most epithelial cells, it is well established that cell proliferation is triggered by increased mechanical tension (Uroz et al, 2018) and inhibited by compression (Alessandri et al, 2013; Delarue et al, 2014; Dolega et al, 2017). MDCK-II cell epithelial monolayer growing under confinement accumulates pressure that inhibits proliferation (Di Meglio et al, 2022). To maintain tissue homeostasis and control cell number, overcrowding results in live cell extrusion in the lumen. It requires sphingosine 1-phosphate G protein coupled receptor signaling and RHO-kinase-dependent myosin contraction. Compared with other types of cell extrusions, this selective process is distinguished by a signaling through SACs (Piezo-1) (Eisenhoffer et al, 2012). Besides, during epithelium growth, epithelial cells spontaneously buckle (Trushko et al, 2020), and cell proliferation is transiently reactivated within the fold. Whereas compressive force-

induced blockage of cell proliferation is dependent on GSK-3$\beta$ (glycogen synthase kinase 3$\beta$) and the $\beta$-catenin transcriptional activator signaling pathways (Song et al, 2017; Di Meglio et al, 2022), reactivation of proliferation within folds correlates with the local reactivation of the mechano-sensing YAP/TAZ pathway through curvature sensing (Di Meglio et al, 2022). Other mechanisms that sense epithelium curvature were described and involve nuclear mechanosensing (Luciano et al, 2021). However, activation of YAP/TAZ remains the most well-described mechanosensing signaling process after mechanical and especially tensile stress (Dasgupta & McCollum, 2019; Cobbaut et al, 2020; Cai et al, 2021). YAP and TAZ transcription coactivators are oncoproteins repressed through their phosphorylation by the tumor suppressor LATS1/2 (large tumor suppressor kinases 1 and 2) controlled by the kinases MST1/2 (macrophage stimulating 1 and 2), mammalian homologs of the Hippo kinase (Dasgupta & McCollum, 2019; Cobbaut et al, 2020; Cai et al, 2021).

### Control of (stem) cell differentiation

Current research highlights a direct relationship between cellular physical property and (stem) cell fate decision, potentially contributing to organ homeostasis and development. Volumetric compression alone induced by either osmotic, mechanical or matrix rigidity controls stemness and intestinal organoid growth; it activates pro-tumoral pathways such as Wnt-activating $\beta$-catenin signaling (Li et al, 2021a).

In bone remodeling, compression forces promote cell differentiation. Prolonged dynamic compression promotes the chondrogenic differentiation of human synovium-derived mesenchymal stem cells in the presence of the transforming growth factor $\beta$3 (Ge et al, 2021), describing a physiologically relevant mechanism of stem cell-based cartilage repair and regeneration. Similarly, long-term mechanical load potentiates the osteogenic differentiation of human BMSCs in collagen microtissues (Song et al, 2017; Li et al, 2020). Compression promotes differentiation of mesenchymal cells and their production of collagen matrix, a process that participates in bone healing and remodeling (Wright et al, 1997; He et al, 2004; Chowdhury et al, 2006; Fitzgerald et al, 2008; Zhong et al, 2013; Liu & Lee, 2014; Takemoto et al, 2015).

Mechanistically, different signaling pathways such as the activation of PI3K/AKT or inhibition of MAPK/ERK signaling promote the compression-induced osteogenic differentiation of BMSCs (Pelaez et al, 2012; Song et al, 2017). In response to compression, the PI3K pathway is activated and increases the $\beta$-catenin expression level which is involved in osteogenic differentiation of BMSCs (Song et al, 2017). Compression also triggers cell differentiation through paracrine regulations between cell types involved in bone regeneration. Cyclic compressive forces, which mimic compression found in bone healing, enhance production and secretion of BMP2 by human BMSCs that stop their migration. Furthermore, secreted BMP2 is required for the expression of the *RUNX2* osteogenic gene (Schreivogel et al, 2019).

Finally, active response to compressive forces is needed for cell differentiation in homeostatic conditions (Nordgaard et al, 2022). In skeletal muscle, the contraction of individual muscle fibers activates p38 MAPK and JNK activation. Osmotic shock and mechanical compression, but not stretching of skeletal muscle cells, selectively activate upstream the ubiquitously expressed but poorly described MAP3K splice form ZAK$\beta$. ZAK$\beta$ is necessary for the proper function of skeletal muscle fibers during contraction; its activation is required to prevent muscle pathology (Nordgaard et al, 2022).

### Promotion of cytoskeleton rearrangement, cell motility, and migration

Compression induces cell signaling to promote cytoskeleton reorganization, cell motility, and migration. In HEK293 cells, compression promotes the activation of RHOA and ROCK signaling critical for actin cytoskeleton remodeling and cell motility (Boyle et al, 2020). Moreover, bronchial primary epithelial cells transition from a nonmigratory to a migratory phenotype upon compression (De Marzio et al, 2021). Fibrous matrix of collagens helps the cells to migrate along the fibers (Hogrebe et al, 2017), production that is favored by compression (Wright et al, 1997; He et al, 2004; Chowdhury et al, 2006; Fitzgerald et al, 2008; Zhong et al, 2013; Liu & Lee, 2014; Takemoto et al, 2015). Compression promotes key cellular processes involved in migration such as formation of lamellipodia and adhesion to extracellular matrix of human BMSCs cultured in collagen matrices (Li et al, 2020; Lim Lam et al, 2021). This process also plays a key role in the function of immune cells or platelet cells that sense compression when they are colliding in blood vessels or in tissues (Toyjanova et al, 2015). Cytoskeleton rearrangement is also occurring as a protective mechanism to physiological high compressive load. Microtubules can bear compressive loads, which is consistent with models for cellular mechanics in which microtubule compression helps to stabilize cell shape by balancing tensional forces within a prestressed cytoskeleton (Wang et al, 1993); this cytoskeleton rearrangement is particularly relevant for cardiomyocytes subjected to constant contractile compressive stresses (Brangwynne et al, 2006).

### Emerging evidence in control of metabolism

Compression modulates the cell metabolism that sustains cell behaviors. Primary human chondrocytes exposed to compression which mimics their natural mechanical load present up-regulation or down-regulation of specific metabolic transcriptional signatures (McCutchen et al, 2017). Nicotinamide metabolism, whose gene signature is down-regulated by compression, is notably essential to produce coenzymes used in glycolysis (McCutchen et al, 2017). The metabolic measurements are, for the moment, too limited to confirm those transcriptomic results. This is an important future field of research as, interestingly, disassembly of actin cytoskeleton network occurs during compression through RHO and ROCK signaling (de Araujo et al, 2014) and could lead to a decrease in glycolysis rate. Indeed, mechanically induced fragmented actin promotes a decreased glycolysis (Park et al, 2020).

### Emerging evidence in control of intercellular communication

A new field of research that could expand within the next years is the study of how compressed cells affect their less compressed or uncompressed neighbors through paracrine signaling (Fig 2C). Orthodontic tooth movement generates both tensile and compressive forces. Periodontal ligament stem cells (PDLSCs) are the main mesenchymal stem cells in periodontal tissues (Jiang et al,

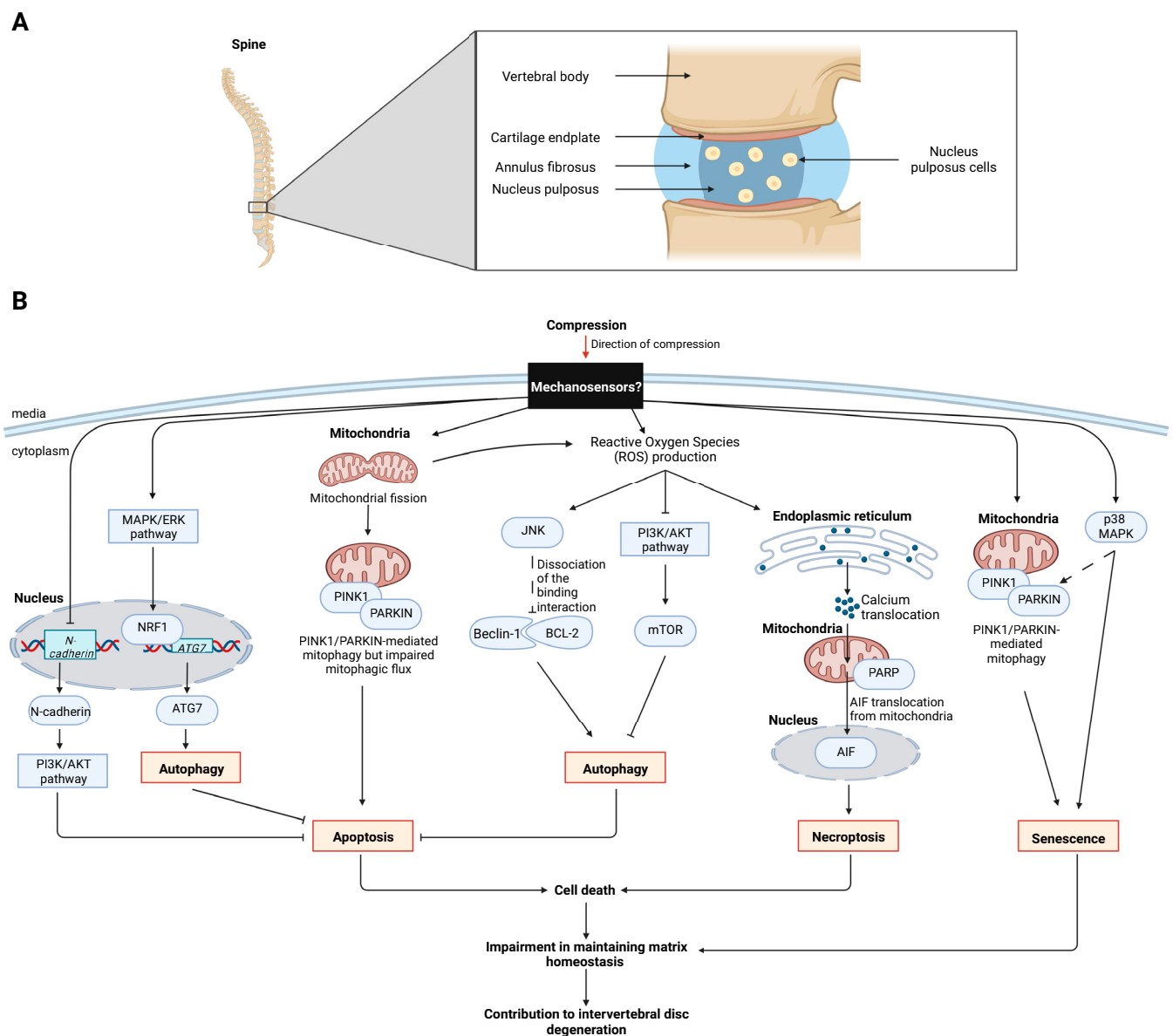

**Figure 3. Effects of compression on cell signaling and death-related cellular outputs of nucleus pulposus cells.**
**(A)** Intervertebral disc structure. **(B)** Effects of compression in nucleus pulposus cells. Signaling pathways involved after compressive stress in nucleus pulposus cells. The activation of signaling pathways tightly regulate the balance between autophagy, cell death (apoptosis, necroptosis), and senescence in cells. The figure compiles data from Ma et al (2013), Li et al (2017, 2018, 2021b), Huang et al (2020), and Lin et al (2021). Undefined modulations are presented as dotted arrows. AIF, apoptosis-inducing factor; ATG7, autophagy-related protein 7; BCL-2, B-cell lymphoma 2; ERK, extracellular signal-regulated kinase; JNK, c-Jun N-terminal kinase; MAPK, mitogen-activated protein kinase; mTOR, mammalian target of rapamycin; N-cadherin, neural cadherin; NRF1, nuclear respiratory factor 1; p38 MAPK, p38 mitogen-activated protein kinase; PARP, poly(ADP-ribose) polymerase; PI3K, phosphoinositide 3-kinase; PINK1, PTEN (phosphatase and TENsin homolog)-induced kinase 1.

2021). Depending on their location in the periodontal tissue, PDLSCs will sense either tension or compression during orthodontic tooth movement (Wise & King, 2008). In vitro compression activates autophagy in PDLCs prompting them to secrete a conditioned medium able to inactivate the AKT signaling of macrophages in a paracrine manner. This causes the polarization of the macrophages into M1 macrophages which next act on bone remodeling and root resorption (Jiang et al, 2021).

## Cell compression in pathological conditions

### Pathological context involving compression forces
Compressive forces are involved in a large number of pathologies. Asthma is a pathology whose development is associated to sensing of compressive forces. Through a paracrine signaling, the compressed primary human bronchial epithelial cells produce and secrete a vasoconstrictor mediator (Endothelin-1) which acts on the primary human airway smooth muscle cells (HSAM cells) to

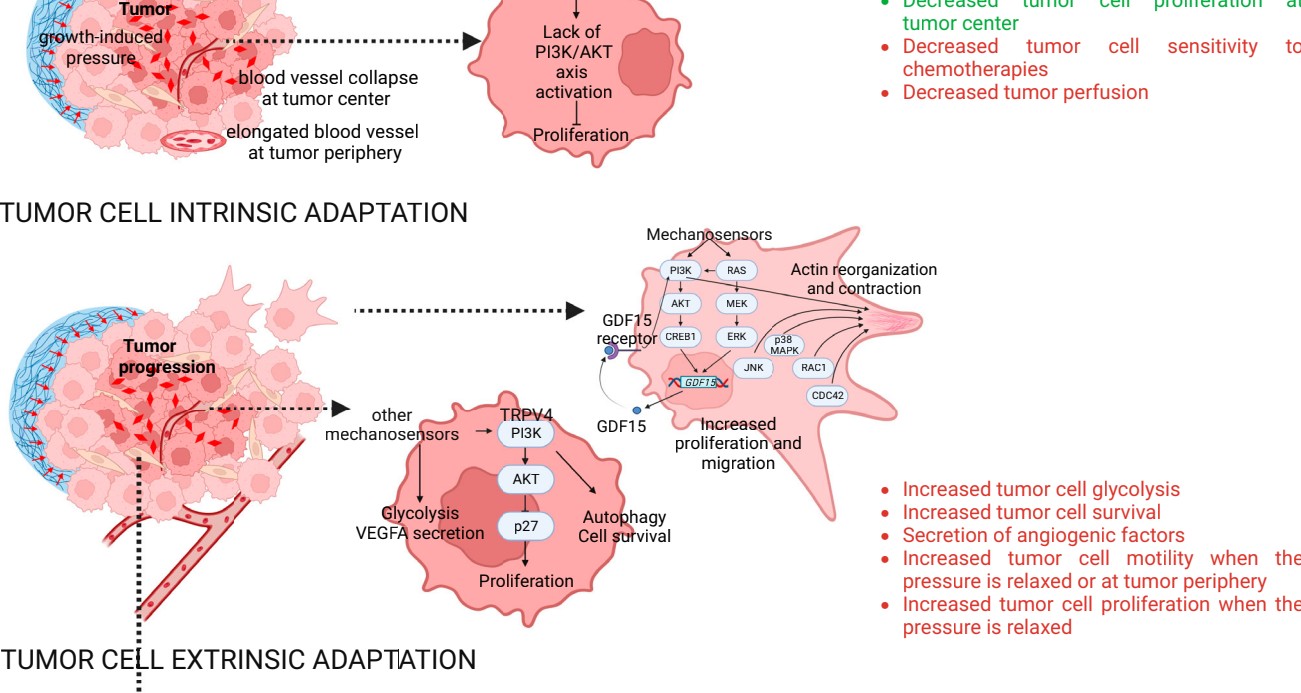
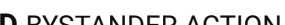
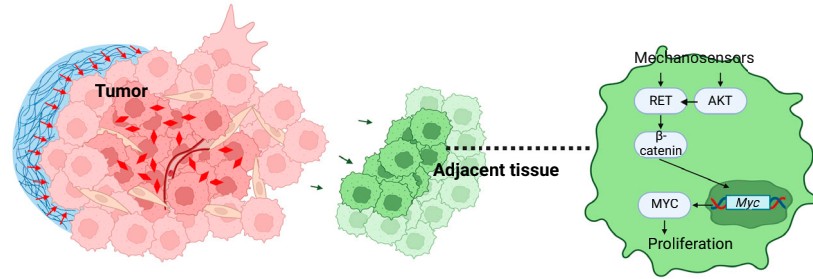

**A** EARLY ACTION OF COMPRESSION

**B** TUMOR CELL INTRINSIC ADAPTATION

**C** TUMOR CELL EXTRINSIC ADAPTATION

**D** BYSTANDER ACTION

**Figure 4. Impact of compressive forces in cancer initiation, progression, and resistance.**
**(A, B, C, D)** This figure is a summary of studies performed in vitro and in vivo in cancer cells (A, B) from different origins such as the brain, breast, and pancreatic cancer, and in cancer-associated fibroblasts (C), and in cancer adjacent tissues (D). For references, see the main text. Text in green refers to positive action of compression, red to negative actions for patient survival. β-catenin, beta-catenin; CDC42, cell division control protein 42 homolog; CREB1, cAMP responsive element-binding protein 1; ENO2, enolase-2; ERK, extracellular signal-regulated kinase; GDF15, growth/differentiation factor 15; HK2, hexokinase 2; JNK, c-Jun N-terminal kinase; MEK, mitogen-activated protein kinase kinase; MYC, c-Myc; p38 MAPK, p38 mitogen-activated protein kinase; PFKFB3, 6-phosphofructo-2-kinase/fructose-2,6-biphosphatase 3; PI3K, phosphoinositide 3-kinase; RAC1, ras-related C3 botulinum toxin substrate 1; RET, ret proto-oncogene; VEGFA, vascular endothelial growth factor A.

increase their proliferation and their basal and histamine-induced contractions. Thus, the HSAM cells are more prone to a future bronchoconstriction leading to a deleterious positive feedback loop for which each bronchoconstriction promotes the next one (Lan et al, 2018). Surface glycoproteins such as MUC family members whose level in the plasma membrane is changed in bronchial

epithelium upon compression further participate in mechano-sensing of compression (Park & Tschumperlin, 2009; Eisenhoffer et al, 2012; Delarue et al, 2017; Massey et al, 2020).

Compression controls both establishment and progression of osteoarthritis, which is a disease characterized by joint inflammation. Thereby, mechanical load induces both human chondrocyte degeneration and cartilage vascular invasion, causing and worsening the pathology, respectively. These effects might be partially mediated by a compression-induced decrease in the activation of TIMP3/TGF-$\beta$1 pathway in the compressed chondrocytes (Zhao et al, 2020).

A prolonged high cerebrospinal fluid pressure in the skull causes intracranial hypertension. Compression of rat cortical neurons that mimics this situation reveals that mitochondrial dysfunction and ER stress occur and cause cell death (Chen et al, 2019). Finding ways to target these causes of neuronal cell death would enable to treat patients suffering from intracranial hypertension.

### Intervertebral disc degeneration induced by excessive compression is caused by massive cell death

The study of the multistep processes leading to pathologies highlights other cellular outputs induced by excessive compressive forces. Intervertebral disc degeneration (IVDD) is a pathology caused by an inappropriate mechanical load on the intervertebral discs leading especially to low back pain (Kang et al, 2020). Intervertebral discs are formed of the peripheral annulus fibrosus, the cartilage endplate, and the central gelatinous nucleus pulpous (NP), which is the most important part for maintaining the homeostasis of these three structures (Fig 3A). Indeed, NP is made of cells producing NP matrix proteins (aggrecan and collagen II) that enable the mechanical functioning of each disc (Han et al, 2017).

Because of an excessive compression, the production of NP matrix proteins is reduced resulting in an impossibility of maintaining the matrix homeostasis (Li et al, 2017). This impairment was explained by the fact that under a mechanical load, the activation of a wide range of signaling pathways in NP cells converge to the control of cell survival/death balance through induction of either autophagy, apoptosis, necroptosis or senescence (Ma et al, 2013; Li et al, 2017, 2018, 2021b; Pang et al, 2017; Huang et al, 2020; Kang et al, 2020; Lin et al, 2021) (Fig 3B). Autophagy is a cellular process that allows the orderly degradation and recycling of cellular components, hence providing a self-promoted nutrient source benefiting cell homeostasis and survival (Poillet-Perez & White, 2019). The compression-induced signaling pathways and their interplays are a matter of future work. It aims to define how to prevent or attenuate the massive cell death of NP cells in patients suffering from IVDD.

### Compression during cancer development—is it promoting or restraining tumor initiation, progression or treatment response?

Unlike IVDD, cancer is not caused by compression, but compressive forces increase during development of solid cancers causing disease progression through various cell modifications (Fig 4). Compressive forces have for origin: (i) the rapid proliferation of cancer cells in a confined environment and, (ii) the accumulation of a non-tumoral environment such as an increase in extracellular matrix content, a remodeling of matrix composition, and matrix swelling with water uptake. Cancer and microenvironment cells sense

compressive forces in a solid tumor. In both cell compartments, compression plays a dual role associated with the evolution of tumors (summarized in Fig 4). Epithelial cells respond to compression by arresting cell proliferation, promoting cell death or extruding cells (in lumens for epithelial cancers), which are protective mechanisms in the early steps of cancer development (Fig 4A); however, cancer epithelial cells adapt to this context and later compression leads to increase cancer cell proliferation, migration, and survival (Fig 4B). Similarly, a compressed microenvironment prevents cancer cell proliferation and tumor vascularization in early steps (Fig 4A), but accelerates cell proliferation, migration, survival to harsh environments in late stages (Fig 4C). Finally, compressive force increase in the tumor is sensed by adjacent tissues and favors tumor initiation (Fig 4D).

**Early action of compression in primary tumors** The thickening of the non-tumoral environment around tumors because of inflammation lead to a compression of the whole tumor (Stylianopoulos et al, 2012; Jain et al, 2014; Northcott et al, 2018; Morikura & Miyata, 2019). In addition, in colon polyps, cancer-associated fibroblasts encapsulate and actively compress epithelial cells. Spontaneous cancer-associated fibroblasts actomyosin contractility is sensed by cancer cells leading to the cytoplasmic relocalization of their YAP proteins, preventing YAP/TAZ transcriptional effect, and reducing epithelial cell growth (Barbazan et al, 2021 Preprint). Hippo pathway containing transcriptional regulators YAP/TAZ can reprogram cancer cells into cancer stem cells and incite tumor initiation, progression, and metastasis. Furthermore, the Hippo pathway crosstalks with morphogenetic signals, such as Wnt growth factors, and is also regulated by RHO and G protein-coupled receptor, cAMP (cyclic adenosine monophosphate), and PKA (protein kinase A) pathways (Dasgupta & McCollum, 2019; Cobbaut et al, 2020; Cai et al, 2021). Class I PI3Ks are known to be upstream activators of YAP/TAZ transcriptional pathway under tensile stress, positioning class I PI3Ks proteins as potential regulators of an essential mechanotransduction signaling (Chronopoulos et al, 2020; Di-Luoffo et al, 2021).

Compressive forces reduce cancer cell proliferation, especially in the solid tumor center (Stylianopoulos et al, 2013; Nam et al, 2019; Rizzuti et al, 2020). With this knowledge, we previously tested and validated in vitro the hypothesis that the compression-induced decreased proliferation of cancer cells reduces the efficiency of chemotherapeutics known to target cycling cells (Rizzuti et al, 2020). It is the case with the use of gemcitabine, reference treatment in pancreatic cancer. Compressive forces modify the blood vessel shape until collapsing its structure, and as a consequence, decreasing the tumor perfusion (Stylianopoulos et al, 2013). By decreasing the tumor perfusion, this would prevent molecular agents to reach the cancer cells, further reducing the drug efficiency (Stylianopoulos et al, 2012) (Fig 4A).

**Tumor cell intrinsic adaptation to compression** Whole transcriptomics analysis reveals that signaling outputs induced by compression converge to control the expression of genes involved in glycolysis in glioblastoma cells (Calhoun et al, 2020). The fact that glycolysis seems to be up-regulated in cancer upon compression is opposite to what is described on chondrocytes (McCutchen et al, 2017). This process might enable cancer cell survival in compressed

environments with low access to nutrients and oxygen (Kim et al, 2019), because of vessel collapse.

Studies converge to show that compression promotes cancer cell motility and their migratory behavior (Pathak & Kumar, 2012). Both the MAPK/ERK pathway, in brain cancer cell lines, and the PI3K/AKT pathway, in pancreatic cancer cell lines, increase the expression of the migration-related *GDF15* (growth/differentiation factor 15) gene which mediates cancer cell migration after their compression (Kalli et al, 2019a, 2019b). In compressed pancreatic cancer cell lines, up-regulation of p38 MAPK and JNK signaling pathways and cytoskeleton remodelers (RAC1 and CDC42) were also shown to promote migration (Kalli et al, 2022). Besides, the compression-induced motility of tumoral cells also depends on the cancer cell type studied. Brain neuroglioma cell lines migrate more under compression than the more aggressive brain glioblastoma cell lines even though both cell lines display an increased motility in response to compression (Kalli et al, 2019b; Calhoun et al, 2020). Molecular mechanism that explains this process possibly involves the regulation of PI3K pathway, as PI3Kα is critical to control metastatic behavior (Ippen et al, 2019; Thibault et al, 2021; Tehranian et al, 2022). Compression also enhances the invasive phenotype of cancer cells by specifically increasing the motility of peripheral tumoral cells (Tse et al, 2012). The increased motility of peripheral tumoral cells upon compressive forces was also found in a 3D experiment for which mouse colon carcinoma cells were encapsulated and grown in alginate capsules (Alessandri et al, 2013).

Compression of cancer cells could further enhance their proliferation when the forces are relaxed. When various epithelial tumor spheroids are grown in softer hydrogels enabling a reduced mechanical confinement in time compared with stiffer hydrogels, increased cell growth occurs and activates SACs such as transient receptor potential vanilloide 4 (TRPV4) that themselves activate the PI3K/AKT axis. This would enable the cytoplasmic relocalization and therefore inhibition of p27 (an inhibitor of cell cycle) resulting in a cell cycle acceleration through a G1 to S phase transition (Nam et al, 2019). This adaptive behavior in response to compression is only unleashed when tumor cells are relieved from their confinement. Indeed, compression without confinement increases the proliferation of colon and glioblastoma cancer cells (Mary et al, 2022).

Interestingly, we and others identify that in the adaptive response to compression of cancer cells, PI3K/AKT activation leads either to cell proliferation, survival or to migration (Kalli et al, 2019a; Nam et al, 2019; Di-Luoffo et al, 2023 *Preprint*), that is either pro-tumoral or pro-migratory. Identifying ways to understand how to block this adaptive response to compression could be a way to limit cancer progression. As a proof of concept, we use a panel of breast and pancreatic cancer cells where the PI3K pathway actively controls cell survival and proliferation. We demonstrate that we can further push the cell fate of cancer cells to trigger cell death under compression if PI3Kα is inactivated by increasing autophagic flux (Di-Luoffo et al, 2023 *Preprint*) data that are reminiscent to the molecular processes described in IVDD (Li et al, 2021b). Increased levels of autophagy in tumor cells promote growth of established tumors and treatment resistance at the progressive stage (Bryant et al, 2019). Combined control of PI3K and autophagy needs to be tested as a novel way to control deleterious adaptive response of cancer cells to compression (Fig 4B).

**Tumor cell extrinsic adaptation to compression** The microenvironment also adapts to compressive forces and is likely to contribute to cancer progression. Compression controls gene expression involved in glycolysis in breast cancer-associated fibroblasts (Kim et al, 2019; Calhoun et al, 2020), that might provide a different metabolite supply to cancer cells; the impact on tumor matrix remodeling is poorly investigated so far (Luo et al, 2022). A recent study revealed that compression increases *VEGFA* gene expression and protein level in breast cancer cells and associated fibroblasts (Kim et al, 2017). Because VEGFA contributes to the formation of new blood vessels (Claesson-Welsh & Welsh, 2013), angiogenesis might be increased upon compression. Nowadays, the impact of compressive forces on tumor vascularization and, consequently, on drug delivery is not fully understood (Fig 4C).

**Bystander action of compression induced by tumors in neighboring normal cells** Growth-induced compressive forces result in an increased compression of the tumor interior that spreads to the boundaries of the tumor (called radial compression) where conjunctive stroma also sense compressive forces. Studies on colon cancers revealed that a growing solid tumor exerts a mechanical pressure on adjacent non-tumoral cells that induces a signaling into cancer-initiating cells causing the formation of adjacent tumoral foci through a pathway involving RET and the β-catenin protein (Fernández-Sánchez et al, 2015; Nguyen Ho-Bouldoires et al, 2022) (Fig 4D).

# Discussion and Future Directions

In homeostasis, compression modifies the signaling pathways and, ultimately, the behaviors of the compressed cells and even indirectly, the adjacent uncompressed cells. Depending on the context, compression either induces or prevents cell proliferation, survival, differentiation, and migration and changes the cell metabolism. These behaviors occur during developmental processes (Xiong et al, 2020).

The impact of compressive forces during development is a vast topic, beyond the field covered in this review and likely to expand. Indeed, some studies show that the fate of stem cells depends on the intensity of the mechanical load: in the presence of an adipogenic medium, rat BMSCs subjected to a small-magnitude stress undergo an osteogenic differentiation, whereas rat BMSCs subjected to a large-magnitude stress undergo an adipogenic differentiation (Song et al, 2017). Stemness is controlled by compression in normal intestinal organoids (Li et al, 2021a). As stemness property is relevant for cancer studies and is notably important in tumor and metastatic dormancy, it is crucial to pursue this line of research in both model organisms and cancer context.

Organ size is thought to be regulated in part by mechanical forces. One hypothesis is that compressive forces increase as the organ grows to reach a threshold inhibiting further organ growth. Inhibition of cell proliferation by compression likely participates in this termination of organ growth (Buchmann et al, 2014). The compressive forces found in adult organs might also regulate cell proliferation to maintain a constant organ size. Indeed, one study suggests that compression might decrease proliferation of skeletal

myoblast cells (Takemoto et al, 2015). How tumor overcomes this physical limitation is a matter of importance.

Compression can drive pathologies (such as IVDD) or buildup during pathology development (such as in cancer). Knowing the selective biochemical signals that are triggered by compression and that contribute to diseases could lead to the discovery of efficient signal-targeted therapies. For example, the studies of IVDD give insight on how to push cancer cells undergoing compression towards cell death. Both the mitochondria-targeted antioxidant MitoQ and the resveratrol molecule could positively impact IVDD in vitro by stabilizing mitochondrial functions and increasing NP matrix synthesis, respectively (Han et al, 2017; Kang et al, 2020). The benefit or detriment of those strategies could be investigated in cancer, where mitochondrial function plays a key role despite the Warburg effect.

To reach the aim of developing therapies that consider the influence of compression in cancer, several key outstanding questions remain.

First, different mechanosensors of compressive forces were identified: compression could change the conformation of ion channels like Piezo-1 and Piezo-2 and activate as a consequence transmembrane proteins like α5β1 integrin, MUC family. Recent studies emphasized that intracellular, molecular, and organelle crowding and nucleus deformation occur and could participate in the starting point of the compression mechanotransduction (Guo et al, 2017; Lomakin et al, 2020; Venturini et al, 2020). However, we still need to identify the selective mechanosensors or the selective mechanism of their activation by compression.

Furthermore, studies should now focus on confirming the implicated pathways in vivo or in refined biomimetic devices. For 3D studies, attention should be paid to the geometry of the produced tissue as it can impact the cell response to compression (Berg et al, 2021). Besides, few in vivo compression methods have been developed. By using magnets subcutaneously inserted close to the mouse colon, Fernández-Sánchez et al deliver an in vivo mechanical pressure, mimicking the one undergone by non-tumoral tissues adjacent to a growing early colon tumor. This study revealed that this compressive force could cause the formation of new tumoral foci in the non-tumoral adjacent tissues (Fernández-Sánchez et al, 2015). Another group, Nia et al used modified cranial windows on mice to recapitulate compressive forces caused by brain tumors (primary and metastatic tumors of the cerebellar cortex and tumors of the cerebellum) (Nia et al, 2020). Combining this device with intravital imaging or other assays could represent a powerful way to study the effect of in vivo compression in cancer.

The signaling pathways induced in compressed cells lead to both cancer and microenvironment cell adaptation by feedback loops (Kalli et al, 2019a, 2019b; Kim et al, 2019; Calhoun et al, 2020). This should be analyzed more comprehensively in various cancer cell types to identify a selective factor critical for the adaptive response to compressive forces. In this topic, we are interested in investigating how PI3K signaling is a generic adaptive response of cancer and microenvironment cells to compression (Di-Luoffo et al, 2021), and in particular, on how autophagy response is a determinant of cancer cell death upon compression and PI3K inhibition (Di-Luoffo et al, 2023 Preprint).

In regenerative medicine, some studies attempt to use compressive forces, or even to combine them with other constraints exerting naturally (e.g., shearing forces), to obtain a model mimicking the human cartilage (Guo et al, 2016). One should be careful to use modulation of compression to treat cancer. If therapies by modifying extracellular matrix could alleviate the pressure within the tumor, they could improve drug efficiency of chemotherapeutic agents (Stylianopoulos et al, 2012; Rizzuti et al, 2020). However, once unconfined, compressed cancer cells might be more aggressive and resist therapies. Fig 4 shows that compressed cancer cells evolve and adapt. They ultimately increase their proliferative and invasive phenotype and their capacity to survive and proliferate in harsh environments (Stylianopoulos et al, 2013; Nam et al, 2019; Rizzuti et al, 2020). This process might trigger selection of selective genetic and epigenetic traits that need to be characterized. Those traits might lead to both cancer progression and to cancer resistance to targeted therapies.

One aspect of the compression-induced cell response that is currently understudied is the relationship between genetic alterations and cancer response to a compressive environment. Emerging evidence on the studied panel of glioblastoma cells suggests that compression-induced cell signaling and its cellular output might depend on the cell aggressivity and their genetic background (Kalli et al, 2019b; Calhoun et al, 2020). We are thus currently developing novel approaches to study this association in an unbiased way.

In conclusion, understanding selective compression-induced effects in cancer is needed for the success of cancer mechanotherapeutics, such as cancer treatment targeting matrix sensing or morphogenetic programs (Sheridan, 2019). Estimating compressive forces in patients is necessary to know more about mechanobiology in cancers and tailor mechanotherapy to each patient. Alleviating or increasing the tumor pressure for patient therapy is still a matter of debate (Leite & Barbosa, 2019). However, as the compressive forces activate selective oncogenic pathways (Di-Luoffo et al, 2023 Preprint), compression could induce novel tumor vulnerabilities that can be targeted by novel emerging mechanotherapies.

## Supplementary Information

## Glossary

Mechanosensor: Protein-sensing mechanical stresses and transmitting this mechanical signal in a biochemical signal such as signaling pathways.

Mechanotransduction: Transformation of a mechanical stress into chemical–biological signals.

BMP (bone morphogenetic protein): Group of growth factors involved in development of tissues including bones. BMP2 is a member of this family.

β-catenin: Member of the catenin protein family which is a subunit of the cadherin protein complex, and which acts as an intracellular

signal transducer. It plays amongst others an important role in the canonical Wnt pathway.

CDC42 (cell division control protein 42 homolog): Member of the Rho family (Ras homolog family) which is a member of the Ras superfamily of small GTPases.

GPCR (G-protein-coupled receptors): Also called seven-transmembrane receptors or heptahelical receptors, they are proteins located in the cell membrane that bind extracellular substances and transmit signals from these substances to intracellular tripartite molecules called a heterotrimeric G protein (guanine nucleotide-binding proteins).

GSK-3 (glycogen synthase kinase 3): Originally identified as a regulator of glycogen metabolism, GSK-3 acts as a downstream regulatory switch for numerous signaling pathways, including cellular responses to WNT, growth factors, insulin, receptor tyrosine kinases (RTK), Hedgehog pathways, and G-protein-coupled receptors (GPCR). Two isoenzymes encoded by two different genes exist (GSK-3$\alpha$ and GSK-3$\beta$).

Hippo/YAP/TAZ pathway: Cell signaling pathway which negatively regulates YAP (Yes-associated protein) and TAZ (transcriptional coactivator with PDZ-binding motif). YAP and TAZ are both transcription coactivators which bind to the TEAD transcription factors (TEAD1–4) and notably promote cell proliferation and survival. When active, the Hippo pathway causes YAP and TAZ nuclear export or degradation. YAP and TAZ activities is also modulated by Hippo pathway-independent mechanisms.

JNK (c-Jun N-terminal kinase): Group of kinases which are members of the mitogen-activated protein kinase (MAPK) family.

MAPK/ERK pathway: Cell signaling pathway activated by growth factors or cytokines. It involves the cascade activation of RAS (GTPase), RAF [serine/threonine kinase, member of mitogen-activated protein (MAP) kinase kinase kinase or MAP3K family], MEK1/2, also called mitogen-activated protein kinase kinases 1 and 2 (kinases with serine/threonine kinase and tyrosine kinase activities) and ERK1/2, also called extracellular signal-regulated kinases 1 and 2 (serine/threonine kinases). ERK1/2 are members of the mitogen-activated protein kinase (MAPK) family of protein.

p38 MAPK: Group of kinases which are members of the mitogen-activated protein kinase (MAPK) family.

PI3K/AKT pathway: Cell signaling pathway in which PI3Ks (phosphoinositide 3-kinases) are activated by several signals (e.g., hormones, growth factors, extracellular matrix) and phosphorylate phosphatidylinositol on 3- hydroxyl group positions of the inositol ring to produce PIP3 [Phosphatidylinositol (3,4,5)-trisphosphate]. Through recruitment of various proteins at the PIP3, the AKT protein is phosphorylated and thus activated to phosphorylate other proteins at serine and threonine sites. This pathway also closely controls actin cytoskeleton rearrangements through RHO and ROCK activation.

RET (rearranged during transfection): Tyrosine kinase receptor which is a subunit of a complex binding to growth factors of the glial-derived neurotropic factor (GDNF) family.

RAC1 (Ras-related C3 botulinum toxin substrate 1): Member of the Rho family (Ras homolog family), which is a member of the Ras superfamily of small GTPases.

RHO (Ras homolog family member): Members of the Rho family (Ras homolog family), which are part of the Ras superfamily of small GTPases. RHOE and RHOA have antagonistics action in actin cytoskeleton regulation.

ROCK (RHO-associated protein kinase): Member of the AGC (PKA/PKG/PKC) family of serine–threonine protein kinases.

RUNX2 (runt-related transcription factor 2): Transcription factor associated with osteoblast differentiation.

SRC (proto-oncogene tyrosine-protein kinase Src): Family of non-receptor tyrosine kinases able to activate, in particular, MAPK/ERK signaling.

Stretched-activated channels (SACs): SACs are described to respond to mechanical forces along the plane of the cell membrane (membrane tension), but not to hydrostatic pressure perpendicular to it.

TIMP3/TGF-$\beta$1 pathway: Cell signaling pathway in which TGF-$\beta$1 (transforming growth factor 1) binds to its cell membrane receptor, leading to the phosphorylation, and thus activation of SMAD2 and SMAD3, which in turn can induce the expression of TIMP3 (tissue inhibitor of matrix metalloproteinase).

WNT: Wnt proteins belong to an evolutionarily conserved family of secreted cysteine-rich glycoproteins. Wnts can activate $\beta$-catenin-dependent canonical Wnt pathway and $\beta$-catenin-independent noncanonical Wnt pathway. A key feature of the canonical Wnt pathway is the regulated degradation of transcription coactivator $\beta$-catenin by the $\beta$-catenin destruction complex which includes glycogen synthase kinase 3$\alpha$ and 3$\beta$ (GSK-3$\alpha$ and GSK-3$\beta$).

# Acknowledgements

A particular thanks to Christophe Trehin, Ecole Normale Supérieure review tutor for advice, and Morgan Delarue and the "CRCT-SigDYN/LAAS-DelarueLab Mechagroup" for their constant discussions on the topic. Our work is funded by Labex Toucan (ANR), Fondation Toulouse Cancer Santé (Mecharesist), Inserm Plan Cancer (PressDiagTherapy followed by MechaEvo), INCA-PLBIO2021, Fondation ARC (ARCPJA2021060003932, ARCPGA2022120005630_6362-3). J Guillermet-Guibert obtained a Fondation Fonroga prize for C Schmitter to perform a master degree on this project. All figures were created using Biorender software.

## Author Contributions

C Schmitter: conceptualization, data curation, formal analysis, investigation, visualization, and writing—original draft, review, and editing.

M Di-Luoffo: data curation, formal analysis, supervision, validation, investigation, visualization, and writing—review and editing.

J Guillermet-Guibert: conceptualization, data curation, formal analysis, supervision, funding acquisition, validation, investigation, visualization, project administration, and writing—original draft, review, and editing.

## Conflict of Interest Statement

The authors declare that they have no conflict of interest.

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
