## [Reviewer comments · Life Science Alliance]

Life Science Alliance

Transducing compressive forces into cellular outputs in cancer and beyond

Céline SCHMITTER, Mickaël Di-Luoffo, and Julie Guillermet-Guibert

DOI: <https://doi.org/10.26508/lsa.202201862>

Corresponding author(s): Julie Guillermet-Guibert, Cancer Research Center of Toulouse

Review Timeline:

Submission Date:	2022-11-29
Editorial Decision:	2023-01-17
Revision Received:	2023-06-02
Editorial Decision:	2023-06-06
Revision Received:	2023-06-13
Accepted:	2023-06-13

Transaction Report:

January 17, 2023

Re: Life Science Alliance manuscript #LSA-2022-01862-T

Dr. Julie Guillermet-Guibert
Cancer Research Center of Toulouse
INSERM UMR-1037
2 av Hubert Curien
Toulouse 31037
France

Dear Dr. Guillermet-Guibert,

Thank you for submitting your manuscript entitled "Transducing compressive stress into cellular outputs in cancer and beyond" to Life Science Alliance. The manuscript was assessed by expert reviewers, whose comments are appended to this letter. We invite you to submit a revised manuscript addressing the Reviewer comments.

Thank you for this interesting contribution to Life Science Alliance. We are looking forward to receiving your revised manuscript.

Sincerely,

B. MANUSCRIPT ORGANIZATION AND FORMATTING:

Reviewer #1 (Comments to the Authors (Required)):

In this interesting and well-written review report, Schmitter and colleagues discuss the consequences of compressive forces on cellular behaviors both in physiological and pathological settings. The authors provide recent and compelling evidence on the importance of the compressive stress and associated signaling pathways in cell/tissue homeostasis and their deregulation in diseases states such as intervertebral disc degeneration (IVDD) and cancers.

This is a timely topic in the emerging field of the mechanobiology. While the impact of extracellular matrix stiffness and contractile forces on signaling and cell biology is well appreciated, that of compressive forces is under-studied and our knowledge remains sparse. Thus this research subject is of great and wide interest and needs attention.

I have few comments and suggestions that may improve some sections:

- 1) Unless I missed it, the glossary has not been included.
- 2) Cells resist to external forces by adjusting the rigidity of their cytoskeleton, including microtubules. The authors could discuss whether any modifications of the dynamic of the microtubules network such as stabilization and/or posttranslational changes of alpha-tubulin has been described in cells under compression constraints.
- 3) Given the important role of YAP and TAZ in mechanosignaling and oncogenesis, the authors should introduce better YAP/TAZ proteins.
- 4) Discussion of potential mechanosensors of the compressive stress should be perhaps introduced sooner for instance during the description of Figure 4B in which the notion of mechanosensors appear.

Reviewer #2 (Comments to the Authors (Required)):

In this review the authors summarize the impact of compression stress in cell biology with special focus on its relevance in pathological conditions. They start by giving an overview on the main cellular responses that are activated upon compression, including proliferation, differentiation, and motility. They also explore metabolism and intercellular communication as two compression-induced cell responses that are gathering importance in the field. Finally they describe how compression stress has an effect in two relevant human pathological conditions i.e IVDD and cancer.

The authors try to write an ambitious review on a complicated and controversial field of research, but unfortunately, I think the manuscript does not reach the minimum quality required to deserve publication.

General issues:

- Several paragraphs should be written in a more logical way, the order does not always make sense and leaves the reader confused and without a clear message. Often it seems that small pieces of information were taken from the literature without putting them into a logical context and flow.
- The manuscript contains grammatical errors and many long sentences in passive voice, that make for difficult reading. The authors also often use the word "could" leaving it unclear whether this is a possibility based on literature or just an idea or suggestion of the authors.
- The authors mention a glossary in their manuscript, but I could not see that anywhere in the manuscript.

Specific comments on figures

- Figures: The figures look overall nice, but don't complement the text very well.

- Fig 1A and text: It does not really become clear why compression is more interesting than shear and tension stress and whether these three can really be kept separate in the different pathologies?
- Fig 1: The division into input (blue) and output (brown) seem not very clear. Mitochondria impairment for example can be a result of compression and could therefore also be brown, right?
- Compared to the other figures, figure 3 seems extremely simple and containing little information.
- Figure 4 and 5 seem overloaded with information, many proteins shown in figure 4 are never mentioned in the text. In figure 5, the right side is more interesting and contains more information than the left side, so it should be relatively bigger.

Specific comments text (just some examples, not exhaustive!):

- Sometimes the authors use past tense, which is confusing. For example, on page 7 "First, the mechanosensors for compression were poorly defined". When was that? Are they nicely defined now?
- The last sentence of the abstract ("...both in physiological and pathological conditions, with a particular interest in human physiopathology.") and the last sentence of the introduction ("Both physiological and pathological conditions will be considered with a particular focus on mammalian cell.") are very similar.
- Abstract and page 4: The authors mention tooth movement. What kind of tooth movement are the authors referring to? Orthodontic treatment for example often leads to major tension stress and not just compression?
- Page 1: An example for a sentence that is extremely difficult to read: "The signaling pathways induced by compression are mostly activated first in the cytoplasm or in the actin cortical cortex and after or concomitantly in other cell compartments such as mitochondria, endoplasmic reticulum or nucleus to induce, depending on the context, different cellular outputs, such as proliferation and cell death."
- Page 2: It seems weird to call it a "cellular output" when bacteria are squeezed to death by more than 20 MPa.
- Page 2: The paragraph on proliferation is written in a confusing way.
- Page 2 and 3 top: "First, the chondrogenic differentiation of human synovium-derived mesenchymal stem cells is inhibited or promoted, respectively, after a compression applied at day 1 or day 21 of a chondrogenic induction using the transforming growth factor β 3, suggesting that the adaptive response to compression consists in promoting differentiation." Is it inhibited, promoted or simply regulated?
- Page 3 and 4: Experimental details such as exact timepoints should not be used in a review.
- Page 5: "Unfortunately, to date, no other crosstalk has been revealed". Why is this unfortunate? What crosstalk are the authors referring to? And what is the "other" crosstalk?
- Page 5: "...environmental constraints such as daily activities". What are those daily activities and why do the authors call them environmental constraints?
- Page 5/6: The whole chapter "Compression during cancer development" is extremely confusing and lacks a logical line of thought. Maybe at least sorting positive and negative aspects of compression in tumors might help. Also, the fact that glycolysis seems to be upregulated in cancers upon compression is opposite to what is described on page 4 for chondrocytes. The authors should at least mention these opposing findings to not leave the reader confused.
- Page 7, another example of a confusing sentence: "In addition to influencing only the establishments or development of a pathology, compression can sometimes influence both."
- Page 8: "The signaling pathways induced in compressed cells could lead to positive or negative feedback loops. Thereby, some molecules produced in response to compression and modifying the cell behavior could also increase or inhibit the compression-induced signaling pathways. This should be analyzed more comprehensively in various cell types to see if it is a common feature in response to compressive forces." This paragraph is very confusing. What is the "common feature"?

Reviewer #1 (Comments to the Authors (Required)):

In this interesting and well-written review report, Schmitter and colleagues discuss the consequences of compressive forces on cellular behaviors both in physiological and pathological settings. The authors provide recent and compelling evidence on the importance of the compressive stress and associated signaling pathways in cell/tissue homeostasis and their deregulation in diseases states such as intervertebral disc degeneration (IVDD) and cancers.

This is a timely topic in the emerging field of the mechanobiology. While the impact of extracellular matrix stiffness and contractile forces on signaling and cell biology is well appreciated, that of compressive forces is under-studied and our knowledge remains sparse. Thus this research subject is of great and wide interest and needs attention.

We thank the reviewer for acknowledging the importance of the topic and we will take into consideration each comments mentioned in the following points:

I have few comments and suggestions that may improve some sections:

1) Unless I missed it, the glossary has not been included.

The glossary has been included accordingly to the reviewer comment.

2) Cells resist to external forces by adjusting the rigidity of their cytoskeleton, including microtubules. The authors could discuss whether any modifications of the dynamic of the microtubules network such as stabilization and/or posttranslational changes of alpha-tubulin has been described in cells under compression constraints.

We thank you for this suggestion. This paragraph was added:

“Cytoskeleton rearrangement is also occurring as a protective mechanism to physiological high compressive load. Microtubules can bear compressive loads, which is consistent with models for cellular mechanics in which microtubule compression helps to stabilize cell shape by balancing tensional forces within a prestressed cytoskeleton (Wang *et al*, 1993); this cytoskeleton rearrangement is particularly relevant for cardiomyocytes subjected to constant contractile compressive stresses (Brangwynne *et al*, 2006).”

3) Given the important role of YAP and TAZ in mechanosignaling and oncogenesis, the authors should introduce better YAP/TAZ proteins.

We thank the reviewer for this suggestion.

“While compressive force-induced blockage of cell proliferation is dependent on GSK-3 β (glycogen synthase kinase 3 β) and the β -catenin transcriptional activator signaling pathways (Song *et al*, 2017; Di Meglio *et al*, 2022), reactivation of proliferation within folds correlates with the local reactivation of the mechano-sensing YAP/TAZ pathway through curvature sensing (Di Meglio *et al*, 2022). Other mechanisms that sense epithelium curvature were

GUILLERMET-GUIBERT Julie
PhD, Inserm DR, SigDYN leader
Tel. : +33(0) 5 82 74 16 52
Email : julie.guillermet@inserm.fr

described, and involve nuclear mechanosensing (Luciano *et al*, 2021). However, activation of YAP/TAZ remains the most well described mechanosensing signaling process after mechanical and especially tensile stress (Dasgupta & McCollum, 2019; Cobbaut *et al*, 2020; Cai *et al*, 2021). YAP and TAZ transcription coactivators are oncoproteins repressed through their phosphorylation by the tumor suppressor LATS1/2 (large tumor suppressor kinase 1 and 2) controlled by the kinases MST1/2 (macrophage stimulating 1 and 2), mammalian homologs of the Hippo kinase (Dasgupta & McCollum, 2019; Cobbaut *et al*, 2020; Cai *et al*, 2021)."

4) Discussion of potential mechanosensors of the compressive stress should be perhaps introduced sooner for instance during the description of Figure 4B in which the notion of mechanosensors appear.

We thank the reviewer for this suggestion and included the scattered knowledge of compression mechanosensors throughout the text.

Examples:

- Compression on chondrocytes triggers Ca²⁺ signaling by activating mechanosensitive ion channels such as Piezo-1 and Piezo-2, two distinct members of **stretch-activated channel** (SAC) family (Coste *et al*, 2010; Han *et al*, 2012; Liu & Lee, 2014).
- Compared to other types of cell extrusions, this selective process is distinguished by a signaling through **stretch-activated channels** (Piezo-1) (Eisenhoffer *et al*, 2012).
- Surface glycoproteins such as MUC family members whose level in the plasma membrane is changed in bronchial epithelium upon compression further participate in mechanosensing of compression (Park & Tschumperlin, 2009; Eisenhoffer *et al*, 2012; Delarue *et al*, 2017; Massey *et al*, 2020).
- When various epithelial tumor spheroids are grown in softer hydrogels enabling a reduced mechanical confinement in time compared to stiffer hydrogels, increased cell growth occurs and activates **stretch activated channels** (SACs) such as TRPV4 (transient receptor potential vanilloide 4) that themselves activate the **PI3K/AKT** axis.

Reviewer #2 (Comments to the Authors (Required)):

In this review the authors summarize the impact of compression stress in cell biology with special focus on its relevance in pathological conditions. They start by giving an overview on the main cellular responses that are activated upon compression, including proliferation, differentiation, and motility. They also explore metabolism and intercellular communication as two compression-induced cell responses that are gathering importance in the field. Finally they describe how compression stress has an effect in two relevant human pathological conditions i.e IVDD and cancer.

The authors try to write an ambitious review on a complicated and controversial field of research, but unfortunately, I think the manuscript does not reach the minimum quality required to deserve publication.

We agree with the reviewer that the topic is difficult to synthesize as, to our knowledge, this field has not been yet gathered in a comprehensive review. In line with all the comments, we have improved our manuscript structure and better delineated the controversies.

GUILLERMET-GUIBERT Julie
PhD, Inserm DR, SigDYN leader
Tel. : +33(0) 5 82 74 16 52
Email : julie.guillermet@inserm.fr

General issues:

- Several paragraphs should be written in a more logical way, the order does not always make sense and leaves the reader confused and without a clear message. Often it seems that small pieces of information were taken from the literature without putting them into a logical context and flow.

We have completely re-written our review and replaced points/ideas in logical way.

- The manuscript contains grammatical errors and many long sentences in passive voice, that make for difficult reading. The authors also often use the word "could" leaving it unclear whether this is a possibility based on literature or just an idea or suggestion of the authors.

We have now clarified what relates to speculations or to acquired data; we focus our discussion on cancer.

- The authors mention a glossary in their manuscript, but I could not see that anywhere in the manuscript.

A glossary was included accordingly to the reviewer comment.

Specific comments on figures

- Figures: The figures look overall nice, but don't complement the text very well.

We are convinced that the figures now better complement the text; New Figure 4 (corresponding to ex-Figure 5) now explains the role of compressive forces in tumor growth and progression.

- Fig 1A and text: It does not really become clear why compression is more interesting than shear and tension stress and whether these three can really be kept separate in the different pathologies?

A paragraph on the interconnection of the three types of mechanical forces and the difficulty to disentangle the selective signals induced by those three types of forces in physiology was added in the paragraph "General knowledge on cell response to mechanical forces" and complement the new Figure 1 on methods to mimic compressive forces *ex vivo*.

"We know that shearing and tensile forces lead to different mechanotransduction signaling especially leading in the activation of different PI3K class of enzymes and PI3K isoforms (reviewed in Di-Luoffo *et al.*, 2021); similarly, this difference in term of signal activation patterns is found between stretching and compressive forces applied on cells (Haudenschild *et al.*, 2009; Takemoto *et al.*, 2015; Nordgaard *et al.*, 2022). This differential pathway activation has physiological implications: healthy cells such as osteoblast precursors and periodontal ligament fibroblasts produce and secrete different proteins depending on which mechanical stress they undergo (He *et al.*, 2004; Zhong *et al.*, 2013; Takemoto *et al.*, 2015). To better understand the importance of mechanical forces in biological processes, there is thus a need to discriminate the selective contribution of compressive forces in activating biochemical pathways. Biological processes *in vivo* are subjected simultaneously to all three types of mechanical forces; disentangling the relative contribution of each physical force in cell processes is thus a complex task. To model *in vitro* the application of compressive forces to mammalian cells in 2D or in 3D, different methods are available. The use of those methodological approaches is expanding in the cell and tumor biology field, described in detail

GUILLERMET-GUIBERT Julie
PhD, Inserm DR, SigDYN leader
Tel. : +33(0) 5 82 74 16 52
Email : julie.guillermet@inserm.fr

in Figure 1B-D.”

- Fig 1: The division into input (blue) and output (brown) seem not very clear. Mitochondria impairment for example can be a result of compression and could therefore also be brown, right?

We thank you for this suggestion. In now new Figure 2 (corresponding to ex-Figure 1), input and output are now separated for a better understanding.

- Compared to the other figures, figure 3 seems extremely simple and containing little information.

Ex-Figure 3 is now merged in new Figure 2 and examples were added to increase the level of information brought by the figure.

- Figure 4 and 5 seem overloaded with information, many proteins shown in figure 4 are never mentioned in the text. In figure 5, the right side is more interesting and contains more information than the left side, so it should be relatively bigger.

Ex-Figure 4 (new Figure 3) was simplified, and Ex-Figure 5 (new Figure 4) was completely modified to better complement the text.

Specific comments text (just some examples, not exhaustive!):

We thank the reviewer for pointing those issues.

- Sometimes the authors use past tense, which is confusing. For example, on page 7 "First, the mechanosensors for compression were poorly defined". When was that? Are they nicely defined now?

We have corrected this accordingly to reviewer comments.

- The last sentence of the abstract ("...both in physiological and pathological conditions, with a particular interest in human physiopathology.") and the last sentence of the introduction ("Both physiological and pathological conditions will be considered with a particular focus on mammalian cell.") are very similar.

We have corrected this issue.

Introduction:

“In this review, we will summarize the current scattered knowledge of compression-induced cell signaling pathways and their subsequent cellular outputs, both in physiological and pathological conditions, such as solid cancers.”

Main text:

“In sum, this review synthetizes by which cell signaling pathways compression induces cellular phenotypes in mammalian cells. We will also discuss the importance of compression in health and disease with a particular focus on cancer.”

- Abstract and page 4: The authors mention tooth movement. What kind of tooth movement

GUILLERMET-GUIBERT Julie
PhD, Inserm DR, SigDYN leader
Tel. : +33(0) 5 82 74 16 52
Email : julie.guillermet@inserm.fr

are the authors referring to? Orthodontic treatment for example often leads to major tension stress and not just compression?

Many thanks for pointing this:

“During orthodontic tooth movement both tensile and compressive forces are generated. Periodontal ligament stem cells (PDLSCs) are the main mesenchymal stem cells in periodontal tissues (Jiang *et al*, 2021). Depending on their location in the periodontal tissue, PDLSCs will sense either tension or compression during orthodontic tooth movement (Wise & King, 2008). *In vitro* compression activates autophagy in PDLSCs prompting them to secrete a conditioned medium able to inactivate the **AKT** signaling of macrophages in a paracrine manner. This causes the polarization of the macrophages into M1 macrophages which next act on bone remodeling and root resorption (Jiang *et al*, 2021).”

- Page 1: An example for a sentence that is extremely difficult to read: "The signaling pathways induced by compression are mostly activated first in the cytoplasm or in the actin cortical cortex and after or concomitantly in other cell compartments such as mitochondria, endoplasmic reticulum or nucleus to induce, depending on the context, different cellular outputs, such as proliferation and cell death."

The paragraph was re-written and is now the following:

“Sensing of compressive forces occurs at various locations in cells (**Figure 2A**). **Mechanotransduction** happens in the plasma membrane or in the actin cortical cortex. Nucleus deformation induces biochemical pathways simultaneously or in a sequential manner. Molecular or organelle crowding in cytoplasm participates to the sensing of compressive forces (Guo *et al*, 2017). Next, compressive forces induce different cellular outputs, ranging from proliferation to cell death (Li *et al.*, 2017; Boyle *et al.*, 2020; Kang *et al.*, 2020; Lin *et al.*, 2021) (**Figure 2A-C**).”

- Page 2: It seems weird to call it a "cellular output" when bacteria are squeezed to death by more than 20 MPa.

We have removed this sentence, as it does not relate to physiological conditions.

- Page 2: The paragraph on proliferation is written in a confusing way.

The whole paragraph was re-written, see p2-3.

- Page 2 and 3 top: "First, the chondrogenic differentiation of human synovium-derived mesenchymal stem cells is inhibited or promoted, respectively, after a compression applied at day 1 or day 21 of a chondrogenic induction using the transforming growth factor β 3, suggesting that the adaptive response to compression consists in promoting differentiation." Is it inhibited, promoted or simply regulated?

We have clarified this sentence:

“Prolonged dynamic compression promotes the chondrogenic differentiation of human synovium-derived mesenchymal stem cells in presence of the transforming growth factor β 3 (Ge *et al*, 2021), describing a physiologically relevant mechanism of stem cell-based cartilage repair and regeneration.”

GUILLERMET-GUIBERT Julie
PhD, Inserm DR, SigDYN leader
Tel. : +33(0) 5 82 74 16 52
Email : julie.guillermet@inserm.fr

- Page 3 and 4: Experimental details such as exact timepoints should not be used in a review.
We modified the text accordingly.
- Page 5: "Unfortunately, to date, no other crosstalk has been revealed". Why is this unfortunate? What crosstalk are the authors referring to? And what is the "other" crosstalk?
We removed this sentence.
- Page 5: "...environmental constraints such as daily activities". What are those daily activities and why do the authors call them environmental constraints?
We removed this sentence.
- Page 5/6: The whole chapter "Compression during cancer development" is extremely confusing and lacks a logical line of thought. Maybe at least sorting positive and negative aspects of compression in tumors might help. Also, the fact that glycolysis seems to be upregulated in cancers upon compression is opposite to what is described on page 4 for chondrocytes. The authors should at least mention these opposing findings to not leave the reader confused.
The whole paragraph was re-written and organised in 4 subparagraphs. They highlight the difference between the early stage of tumor growth where compression prevents tumor growth and the progressive stage when the tumor cell intrinsically and extrinsically adapts leading to deleterious impact of compression. The difference between cancer cell and chondrocyte response on metabolic gene signatures was highlighted:
"The fact that glycolysis seems to be upregulated in cancer upon compression is opposite to what is described on chondrocytes (McCutchen *et al*, 2017)."
- Page 7, another example of a confusing sentence: "In addition to influencing only the establishments or development of a pathology, compression can sometimes influence both."
This sentence was removed.
- Page 8: "The signaling pathways induced in compressed cells could lead to positive or negative feedback loops. Thereby, some molecules produced in response to compression and modifying the cell behavior could also increase or inhibit the compression-induced signaling pathways. This should be analyzed more comprehensively in various cell types to see if it is a common feature in response to compressive forces." This paragraph is very confusing. What is the "common feature"?
This paragraph was removed accordingly to reviewer comment.

June 6, 2023

RE: Life Science Alliance Manuscript #LSA-2022-01862-TR

Dr. Julie Guillermet-Guibert
Cancer Research Center of Toulouse
INSERM UMR-1037
2 av Hubert Curien
Toulouse 31037
France

Dear Dr. Guillermet-Guibert,

Thank you for submitting your revised manuscript entitled "Transducing compressive stress into cellular outputs in cancer and beyond". We would be happy to publish your paper in Life Science Alliance pending final revisions necessary to meet our formatting guidelines.

- please upload your figures as single files
- the full name (middle name as initials) of each author should be given on the title page
- please add an Author Contributions section to the manuscript text, right after the Acknowledgements section
- please add your main legends to the main manuscript text after the References section
- in the legend for Figure 3B, you say it was adapted from multiple sources. Is the figure itself adapted from another figure published elsewhere? If so, then this needs to be clear and permission is needed to adapt the figure from that publication. If you mean that this figure just compiles information from the listed References, but is a unique figure that you created, then that is fine.

Formatting points:

- there should be an Introduction section, and it seems the "General knowledge on cell response to mechanical forces" section can be renamed as the Introduction
- please rename final section to Discussion and Future Directions

A. FINAL FILES:

B. MANUSCRIPT ORGANIZATION AND FORMATTING:

Sincerely,

June 13, 2023

RE: Life Science Alliance Manuscript #LSA-2022-01862-TRR

Dr. Julie Guillermet-Guibert
Cancer Research Center of Toulouse
INSERM UMR-1037
2 av Hubert Curien
Toulouse 31037
France

Dear Dr. Guillermet-Guibert,

Thank you for submitting your Review entitled "Transducing compressive forces into cellular outputs in cancer and beyond". It is a pleasure to let you know that your manuscript is now accepted for publication in Life Science Alliance. Congratulations on this interesting work.

DISTRIBUTION OF MATERIALS:

Again, congratulations on a very nice paper. I hope you found the review process to be constructive and are pleased with how the manuscript was handled editorially. We look forward to future exciting submissions from your lab.

Sincerely,
